# General and COVID-19-Related Mortality by Pre-Existing Chronic Conditions and Care Setting during 2020 in Emilia-Romagna Region, Italy

**DOI:** 10.3390/ijerph182413224

**Published:** 2021-12-15

**Authors:** Nicola Caranci, Chiara Di Girolamo, Letizia Bartolini, Daniela Fortuna, Elena Berti, Stefano Sforza, Paolo Giorgi Rossi, Maria Luisa Moro

**Affiliations:** 1Regional Health and Social Care Agency, Emilia-Romagna Region, 40127 Bologna, Italy; nicola.caranci@regione.emilia-romagna.it (N.C.); letizia.bartolini@ausl.re.it (L.B.); daniela.fortuna@regione.emilia-romagna.it (D.F.); elena.berti@regione.emilia-romagna.it (E.B.); stefano.sforza@regione.emilia-romagna.it (S.S.); marialuisa.moro@regione.emilia-romagna.it (M.L.M.); 2Epidemiology Unit, Azienda Unità Sanitaria Locale-IRCCS di Reggio Emilia, 42122 Reggio Emilia, Italy; paolo.giorgirossi@ausl.re.it

**Keywords:** COVID-19, mortality excess, chronic conditions, care setting, Italy

## Abstract

In 2020, the number of deaths increased in Italy, mainly because of the COVID-19 pandemic; mortality was among the highest in Europe, with a clear heterogeneity among regions and socio-demographic strata. The present work aims to describe trends in mortality and to quantify excess mortality variability over time and in relation to demographics, pre-existent chronic conditions and care setting of the Emilia-Romagna region (Northern Italy). This is a registry-based cross-sectional study comparing the 2020 observed mortality with figures of the previous five years by age, sex, month, place of death, and chronicity. It includes 300,094 deaths in those 18 years of age and above resident in the Emilia-Romagna region. Excess deaths were higher during the first pandemic wave, particularly among men and in March. Age-adjusted risk was similar among both men and women (Mortality Rate Ratio 1.15; IC95% 1.14–1.16). It was higher among females aged 75+ years and varied between sub-periods. Excluding COVID-19 related deaths, differences in the risk of dying estimates tended to disappear. Metabolic and neuropsychiatric diseases were more prevalent among those that deceased in 2020 compared to the deaths that occurred in 2015–2019 and therefore can be confirmed as elements of increased frailty, such as being in long-term care facilities or private homes as the place of death. Understanding the impact of the pandemic on mortality considering frailties is relevant in a changing scenario.

## 1. Introduction

In Italy, the year 2020 recorded the highest number of deaths since 1917 (more than 700,000) with a strong impact on the country’s demographics [1]. At the national level, mortality was approximately 10% higher than the average figures of previous years; this unprecedented increase was mainly due to the high contribution of deaths from COVID-19, which placed the country among those with the highest COVID mortality rates globally [2]. In 2020 the pandemic showed two main peaks in Europe, one in March/May and the second in September/November. Italy was not an exception, considering that it has been among the first European countries where COVID-19 has appeared, with the first peak occurring in March and the second in November, later than in most other European countries [3]. In Italy, we can therefore distinguish three periods; March–May 2020: with a quick and steep rise in cases and deaths and strong concentration in the Northern regions; June–September: with limited epidemic spread; and October-December: with a new rise of cases and deaths country-wide [4]. It is worth noting that the case fatality rate quickly decreased after the first peak [5], thanks to a higher diagnostic capacity leading to the detection of more asymptomatic and pauci-symptomatic cases. Indeed, in the first weeks of the pandemic testing capability was limited and therefore tests were done mostly on symptomatic people presenting to emergency rooms [6]. Later on, the volume of performed tests increased, allowing the diagnosis of pauci-symptomatic and asymptomatic contacts [7].

Before the vaccination campaign, the impact on mortality of the pandemic mainly depended on the virus spread, the personal characteristics of the affected people (age, sex, pre-existing conditions and pathologies, social characteristics), and the ability of the health service to provide care. The role of pre-existing conditions and pathologies has been investigated in many studies; in the European context, several of these have been conducted by analysing subjects with Corona Virus Disease (COVID-19). The main risk factors and diseases identified as associated with COVID-19 mortality are diabetes, obesity, chronic obstructive pulmonary disease (COPD), kidney disease, dementia, cancer, and cardiovascular disease [8,9,10,11,12,13,14,15]. In addition, an Italian study showed that people living in socially and demographically deprived areas have a higher risk of dying from COVID-19 [16]. Finally, mortality was also influenced by the ability of the health care system and elderly care facilities to respond to the epidemic [17], both in preventing the spread of infection and in providing effective care. 

The present work aims to describe the trends in mortality during the year 2020 and to quantify the excess of mortality in 2020 compared to that observed in the previous five-years period, as well as excesses variability over time and in relation to demographics, the care setting and pre-existent chronic conditions in the Emilia-Romagna region.

## 2. Materials and Methods

### 2.1. Study Design

This is a registry-based cross-sectional study comparing the overall mortality observed in 2020 to the expected one (observed in the previous five years), and by age, sex, month, and place of death. The prevalence of chronic conditions pre-existing the deaths are also compared among the deaths that occurred in 2020 and those occurred in 2015–2019. 

To explore if there were changes in causes of death other than COVID-19 in 2020, we also analysed mortality after excluding COVID-19 related deaths from the casualties occurred in 2020.

### 2.2. Mortality Data and Population Denominators

Mortality data were collected from the 1 January 2015 up to 31 December 2020. We included all deaths that occurred during these six years and referred to people aged 18 and over residing in the Emilia-Romagna region (Northern Italy) on 1 January of each year. The count of all deaths was obtained from the regional health insurance card system (RHICS) data, which guarantees an update of the vital status information and allows to reliably identify deaths within two weeks from the occurrence. Data on deaths directly attributable to COVID-19 were obtained from the regional COVID-19 notifications system [18]. Starting from 22 February 2020, this system collects demographic and clinical information on subjects diagnosed with the SARS-CoV-2 infection, including the infection status (still sick, recovered, or dead) [5]. We take account of all the deaths that occurred in people currently infected or infected within the 30 previous days.

All deaths identified through the two data sources, RHICS and COVID-19 notifications, were linked to the regional population and health databases by an anonymous numeric key, in agreement with European GDPR and regional regulation on personal data. 

Population denominators as of 1 January in each of the six years were obtained from the Regional Bureau of Statistics (RBS).

Counts of total deaths and on those directly attributable to COVID-19 were stratified according to sex and age group (18–74, 75+). The period of the year was also considered, classified as January–February (pre-pandemic period), March–May (first pandemic peak period), June–September (low incidence pandemic period), and October–December (second pandemic peak period).

### 2.3. Classification of Chronic Conditions and Care Setting

#### 2.3.1. Selection of Chronic Conditions

The prevalence of chronic conditions among the deceased was retrospectively investigated, and a list of thirty-one relevant chronic diseases were selected through an identification algorithm [19]. Each disease was considered present when at least one of the following criteria was satisfied (Appendix A): clinical modification of the ninth revision of the International Classification of Diseases (ICD-9-CM) diagnosis reported in the hospital discharge database in the previous four years; Anatomical Therapeutic Chemical (ATC) drug classification system codes of prescriptions in the previous two years; specific chronic condition payment exemption codes effective in each year. For dementia cases only, specific home health care episodes and residential or semi-residential care, provided over the previous five years, were also considered. The chronic conditions were classified into seven groups (Appendix A).

#### 2.3.2. Care Setting

We included the information about the care setting (place of death) in the last week before the death and we classified it according to the six following groups: 1. Hospital (excluding long-term hospitalizations); 2. Community /Long-term Hospital; 3. Hospice; 4. Long-term Care Facilities (LCF); 5. Home Care; and 6. None of the above. In case of multiple care settings during the last week, we considered highest intensity of care (e.g.,: Hospital, Community/Long-term Hospital, etc.).

### 2.4. Statistical Analysis

To evaluate the excess mortality, the number of deaths that occurred in 2020 (observed) were compared with those of the previous five years (expected), and calculated:absolute daily number of overall and COVID-19 deaths, comparing 2020 and 2015–2019 mean, by sex and age group;relative frequencies of 2020 deaths by sex, age group (18–74, 75+), period, pre-existent chronic conditions, and care setting;mortality excesses (observed minus expected) and number of COVID-19 deaths, by sex, age group and period;estimates of Mortality Rate Ratios (MRR: *exp*(*β*_1_)) and 95% confidence intervals, through Poisson models, comparing deaths (*Y*) counted in 2020 with those counted in 2015–2019 (*X*_1 *=*_ dummy for period: [2016–2019 = 0, 2020 = 1]), adjusted for age (five- year classes: vector [*X*_2_]), by sex, age group, care setting (stratifying variables). Population at 1 January of each of the six years was included in models as offset (*P*):
*Y_l_|X*_1,_*X*_2_* ~ Poisson (µ_l_); µ: expected value and the variance of Y_l_, l: care setting;**_i,j_µ_l_ = exp (_i,j_β_1l_ * x_1_ + [_i,j_β_2l_]*[x_2_])*P_i,j_;**i: sex, j: age class*(1)

estimates of Prevalence Ratios (PR: *exp*(*β_d_*)) and 95% confidence intervals, through log-binomial models and their coefficients in explaining probability (*p*) of dying in 2020 (*Y:* death in 2020) by means of dichotomous variables (*X_d_*) indicating pre-existing chronic conditions (*d*), adjusted for age (five years class: vector [*X*_8_]), by sex, age group (stratifying variables):

*Y|X*_1,_ …, *X*_7_, [*X*_8_] *~ Log-binomial*(*p*);
(2)Y|X1…,X7,[X8]~Log-binomial(p);pi,j=exp{(∑d=17βi,jd∗xi,jd)+[βi,j8]∗[xi,j8]}; d: disease, i: sex, j: age class

The models (Appendix A) were applied considering both the overall mortality and the mortality not directly attributable to COVID-19. 

Data management and statistical analyses were carried out by SAS 9.3 software (SAS Institute, Cary, NC, USA) and Stata 15.1 (StataCorp LLC, Lakeway Drive, TX, USA).

## 3. Results

### 3.1. Deaths in Emilia-Romagna Region

Among the adult population (≥18 years) resident in the region on the 1 January of each year, there were 300,094 deaths (Appendix A). In 2020 there were 58,162 deaths, with a 20.2% excess (9776 deaths) according to the mortality figures in the period 2015–2019. Among the total 2020 events, there were 7955 COVID-19 related deaths from 28 February onward according to the COVID-19 information system. 

Figure 1 shows that in January and February 2020 there were less deaths than the average number of events registered in the corresponding months of 2015–2019; in the periods March-May and September–December 2020, observed figures outnumbered the expected ones and the COVID-19 deaths emerged as the main driver of the excess mortality. In the summer months, the COVID-19 related deaths decreased substantially and virtually disappeared. The first peak was stronger in males, in both age groups, and daily mortality dramatically peaked in late March, when the COVID-19 related deaths alone equaled the expected overall mortality (Figure 1). In summer and autumn, the number of deaths showed a much more similar pattern in males and females (Table 1). 

The crude excess mortality was greater among men (22.2%) than women (18.4%), and among the elderly; the share of the excess attributable to COVID-19 related deaths was instead larger in the age group 18–74 years. Excess mortality peaks occurred when COVID-19 related mortality was highest (Table 1) and during the second peak they were almost the same as COVID-19 deaths.

### 3.2. Mortality Figures

In 2020, the age-adjusted risk of death was 15% higher than in the period 2015–2019 (MRR 1.15; IC95% 1.14–1.16) among both men and women. Female risk was noticeably higher among the 75+ than 18–74 age group. The risk of dying was highest during the first epidemic peak among men (March–April MRR 1.48; IC95% 1.44–1.51) and during the second epidemic peak among women (October-December MRR 1.25; IC95% 1.22–1.28), for whom a slight excess persisted even during the summer months (May-September MRR 1.06; IC 95% 1.03–1.08). Compared to the period 2015–2019, in 2020 more deaths occurred at home or in LCF; women reached the highest MRR, particularly in LCF, and men had the highest but unstable MRR corresponding to home care. On the contrary, fewer deaths were registered in other care settings, such as hospices and community or long-term hospitals, among both men and women. The same analyses carried out excluding the COVID-19 related deaths revealed that there were not substantial differences in the risk of dying between 2020 and 2015–2019 and that the shape of non-COVID-19 deaths risks across the care settings mirrored overall mortality trends, apart from deaths in hospital, which were lower in 2020 than in 2015–2019, and in LCF for women, which was higher in 2020 than in 2015–2019 (Table 2).

### 3.3. Prevalence of Chronic Pre-Existent Conditions

The prevalence of selected groups of chronic conditions among the deceased changed between 2015–2019 and 2020 (Appendix A).

After accounting for the effect of age and the concurrent presence of the other groups of chronic diseases, metabolic and neuropsychiatric diseases were still more prevalent among those deceased in 2020 compared to the deaths that occurred in 2015–2019. On the contrary, the prevalence ratios were smaller than one for cardiovascular diseases (PR ≈ 0.9). The prevalence of respiratory disease was lower in 2020 among deceased older than 75 years, while it was higher, even if not significantly, among deceased younger than 75 years, in both sexes. The prevalence ratio of cancer between deaths occurred in 2020 and those registered previously shows differences: it was lower in males below 75 years of age and higher in females over 75 years of age (Figure 2a). When excluding the COVID-19 related deaths that occurred in 2020, the prevalence ratios did not substantially change, except for the higher value for prevalence ratio of cancer in males over 75 years of age (Figure 2b).

## 4. Discussion

In 2020, for the entire Emilia-Romagna region, almost 9000 deaths more than expected were recorded, compared to less than the average 50,000 annual deaths that occurred in the previous five-years, when they remained substantially stable. Deaths from COVID-19 during 2020 were close to 8000. The intensity of mortality corresponds to an average risk increase of 15% in 2020 compared to the previous five years, both among men and women after accounting for the effect of age. 

The trend and intensity in overall excess mortality was similar to the extent of deaths among COVID-19 cases, although this does not fully cover the exceptional increases observed in 2020. This is due to a lower number of COVID-19 deaths, compared to the excess mortality during the first peak period and during the summer. It is possible that some COVID-19 deaths underwent undetected during the firsts weeks of the pandemic because the diagnostic ability of the health system was not optimal, and it was very difficult to perform swabs at the patients’ homes. Furthermore, it is worth highlighting that in January and February 2020, mortality was below the average, and this may have resulted in an underestimation of the expected deaths in the period March–May because of a potential compensatory excess of deaths. The same explanation does not apply to the difference in COVID-19 related deaths and excess deaths during the summer: in this period the diagnostic services and the contact tracing system were not under pressure and undiagnosed cases quickly diminished, as shown by the decrease in case fatality rate [5]. Further research is needed to understand if the summer excess was due to the effects of the long-COVID syndrome among COVID-19 survivors or if it was related to the lockdown consequences on health and health care provision, which was heavily curbed during the first phases of the pandemic [20]. Furthermore, an excess of mortality slightly higher than the number of COVID-19 related deaths, together with a prolonged excess in the summer period, suggest that the harvesting effect of COVID-19, i.e., bringing forward the death by a few days or weeks of people that would have died in any case, is small, if any, even in a country with a very old population, such as Italy. 

Men seemed to be the most affected population group during the first period while women were towards the end of the year. Moreover, female over-mortality was less intense but more prolonged, particularly in the elderly, and was persistent in LCF, also excluding mortality for COVID-19. In Emilia-Romagna, what can be seen in the comparison of Italy with the rest of Europe (Eurostat, 2021) is even more pronounced: an intensity greater in the first weeks of the epidemic and lower in October-December, when another strong wave hit harder those regions initially spared by the epidemic, and continued until the beginning of 2021 [21], with a displacement of the greater impact on mortality that may also have occurred due to the permanence of frail people. Considering the whole year, the relative risk increases, adjusted for age, are not dissimilar between men and women.

The risk of dying at home was higher, both in relation to all deaths, including those from COVID-19, for which it can be hypothesized a greater difficulty in receiving assistance during epidemic peak periods, and for deaths not directly attributable to COVID-19, for which there may have been self-limitations to access to healthcare assistance needs for other reasons than COVID-19 infection [20]. The same phenomenon was observed in England, Wales and Scotland, and explained as not necessarily due to a lack of healthcare, but to a reduced access to hospital, perhaps due to people’s preferences [22,23]. The risk of dying in LCF was also higher. The relationship between COVID-19 mortality and the spread of infection in LCF has been highlighted in the United Kingdom [24] and also in Italy [17,25], particularly for the elderly and people with dementia [26]. Excluding deaths from COVID-19, the excesses in death risk found in LCF tend to shrink, except for the persistence of an excess among elderly women. 

The analysis of the pre-existing chronic conditions, accounted for independently, shows a picture consistent with the literature: we recorded a higher prevalence of metabolic diseases among the deceased in 2020 than those subject that died in 2015–2019; similarly, an increase in the hospitalisation and mortality for diabetes has been largely documented [13]. Neuropsychiatric diseases too were shown to be more prevalent in the literature, with reference to dementia [8], schizophrenia [27], and epilepsy [28]. On the contrary, neoplasms among non-elderly men and cardiovascular diseases among men and women in the two age groups analysed seem to show a defect, despite an impact on COVID-19 prognosis has been also documented for these diseases [18,29,30]. Unfortunately, it is impossible to assess if and how the risk of dying changed during the pandemic compared with previous years according to the presence of pre-existing chronic conditions, because we cannot currently calculate the share of the total population affected by the selected chronic conditions. 

This study is based on data covering an entire year for an Italian region with more than four million residents and among the most affected by the COVID-19 epidemic. Unlike many studies published in the European context, this work considered the characteristics of all deceased persons and not only those deceased from COVID-19, who are prone to different detection methods from one country to another. Nevertheless, associations with some factors related to mortality were generally consistent with those that emerged from the studies including only COVID-19 cases. 

As mentioned above, the study has some limitations: our results could be affected by underestimation, in particular for deaths attributable to COVID-19 in the first period, because of an incomplete notification. Another main limitation is the gap in identifying the total population affected by the selected chronic conditions, which makes it impossible to estimate relative risks and obliged us to estimate the probabilities of having those conditions among the dead. Our analyses currently cover the year 2020, leaving out the observation of the second epidemic phase. Moreover, socio-economic status has not been included in explanatory factors, although it can act as a distal determinant and can be a useful element of knowledge to support prevention strategies [10]. Lastly, the study considered only multiple independent and non-exhaustive pathological conditions without considering multi-morbidity that can modify people’s susceptibility [12].

## 5. Conclusions

In 2020 the impact of COVID-19 pandemic on mortality in Emilia-Romagna was higher in the elderly and during the first epidemic peak. It remained almost parallel to deaths for COVID-19 and, by the second peak, deaths for COVID-19 accounted for most of the excess mortality. Excesses were heterogeneous among periods and between men and women, with the latter more hardly hit in the second epidemic period. Our results support the hypothesis that being at home is associated with higher risk, such as being in an LCF, which were exceptionally overwhelmed. Metabolic and neuropsychiatric diseases have been confirmed as elements of increased frailty. 

Monitoring mortality trends, also including or excluding deaths from COVID-19, remains an important element of surveillance. Looking ahead, preexisting multi-morbidity and socio-economic status are elements to take into account. Moreover, considering the cause of death will be relevant to understand how the impact of the COVID-19 has rearranged their distribution. It will also be useful to extend the time period to appreciate similarities and differences between first and subsequent waves, when new variants of the virus spread and the vaccination campaign was implemented.

## Figures and Tables

**Figure 1 ijerph-18-13224-f001:**
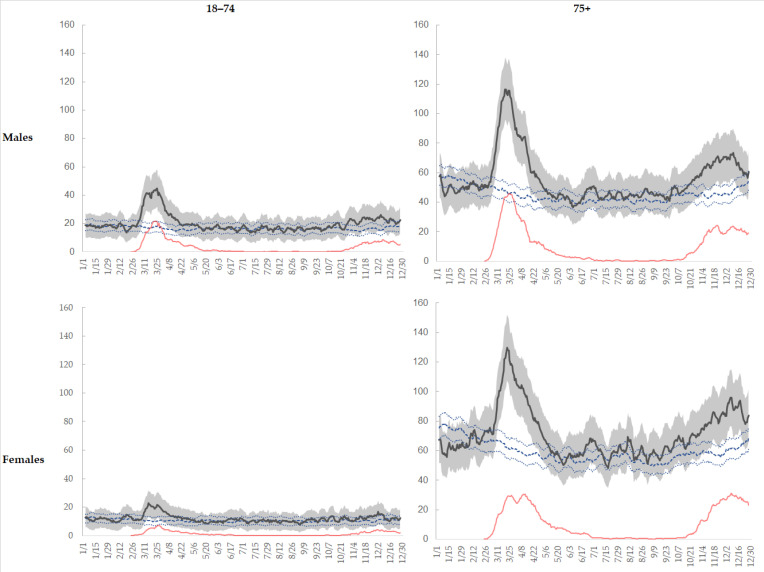
Number of daily total deaths (weekly moving average; on X axis month/day of month) by sex and age classes, 2020 (solid line) vs. 2015–2019 (dots) and number of daily COVID-19 deaths (weekly moving average, red line) by sex and age group, Emilia-Romagna.

**Figure 2 ijerph-18-13224-f002:**
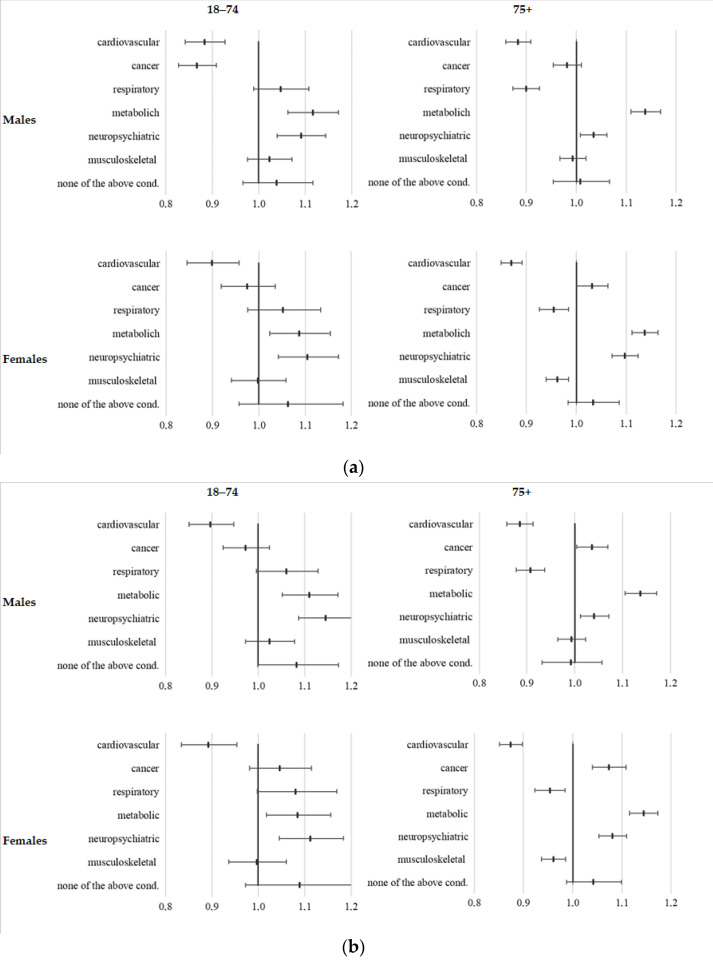
(**a**) Prevalence ratios of pre-existing chronic conditions among deceased, 2020 vs. 2015–2019, by sex and age group, Emilia-Romagna. (**b**) Prevalence ratios of pre-existing chronic condition among not COVID-19 deceased, 2020 vs. 2015–2019, by sex and age group, Emilia-Romagna.

**Table 1 ijerph-18-13224-t001:** Observed total deaths (2020), difference with expected deaths (2015–2019) and composition percent, observed COVID-19 deaths and ratio—on differences—percent, by sex, age group and period, Emilia-Romagna.

			Males					Females				
			Obs. Deaths (OBS)	OBS-Expected (ΔOE)	ΔOE/Expected %	COVID-19 Deaths (COV)	COV/ ΔOE %	Obs. Deaths (OBS)	OBS-Expected (ΔOE)	ΔOE/Expected %	COVID-19 Deaths (COV)	COV/ ΔOE %
Period age-group	January–February	18–74	1078	−16	−1.5	0	0.0	698	−18	−2.5	0	0.0
		75+	3021	−197	−6.1	2	−1.0	3898	−379	−8.9	1	−0.3
		Tot.	4099	−213	−4.9	2	−0.9	4596	−398	−8.0	1	−0.3
	March–May	18–74	2358	836	54.9	693	82.9	1256	291	30.2	245	84.2
		75+	6422	2337	57.2	1717	73.5	7753	2295	42.0	1572	68.5
		Tot.	8780	3173	56.6	2410	75.9	9009	2586	40.3	1817	70.3
	June–September	18–74	1983	1	0.1	35	3500	1264	30	2.4	16	54.1
		75+	5410	398	7.9	101	25.4	7209	657	10.0	165	25.1
		Tot.	7393	399	5.7	136	34.1	8473	687	8.8	181	26.4
	October–December	18–74	1923	398	26.1	388	97.5	1127	148	15.2	163	109.8
		75+	5578	1294	30.2	1330	102.8	7184	1701	31.0	1527	89.8
		Tot.	7501	1692	29.1	1718	101.5	8311	1849	28.6	1690	91.4
	Age-class	18–74	7342	1219	19.9	1116	91.6	4345	451	11.6	424	94.1
		75+	20,431	3833	23.1	3150	82.2	26,044	4273	19.6	3265	76.4
		Total	27,773	5051	22.2	4266	84.5	30,389	4724	18.4	3689	78.1

**Table 2 ijerph-18-13224-t002:** Mortality rate ratios (MRR and 95% Confidence Intervals—95% CI) for general and not COVID-19 deaths: 2020 vs. 2015–2015 mean by sex, age group and care setting. Emilia-Romagna.

	Care Setting	General Mortality				Not COVID-19 Mortality			
		18–74		75+		18–74		75+	
		MRR	(95% CI)	MRR	(95% CI)	MRR	(95% CI)	MRR	(95% CI)
Male	Hospital (excluding long-term hospitalizations)	1.25	(1.21–1.30)	1.20	(1.17–1.22)	0.98	(0.95–1.00)	0.89	(0.87–0.91)
	Community/Long-term Hospital	0.65	(0.56–0.75)	0.73	(0.69–0.78)	0.61	(0.53–0.71)	0.61	(0.53–0.71)
	Hospice	0.91	(0.84–1.00)	0.95	(0.89–1.02)	0.90	(0.82–0.98)	0.90	(0.82–0.98)
	Long-term Care Facilities	1.28	(1.06–1.55)	1.28	(1.20–1.36)	1.03	(0.84–1.27)	1.07	(1.00–1.15)
	Home Care	2.21	(1.11–4.37)	1.36	(1.00–1.86)	2.02	(1.00–4.09)	1.10	(0.79–1.54)
	None of the above	1.15	(1.10–1.20)	1.21	(1.18–1.25)	1.13	(1.09–1.18)	1.18	(1.15–1.21)
	Total	1.15	(1.12–1.18)	1.15	(1.13–1.17)	0.98	(0.95–1.00)	0.97	(0.96–0.99)
Female	Hospital (excluding long-term hospitalizations)	1.05	(1.01–1.11)	1.13	(1.11–1.15)	0.87	(0.82–0.91)	0.89	(0.87–0.91)
	Community / Long-term Hospital	0.55	(0.46–0.67)	0.72	(0.68–0.76)	0.52	(0.43–0.63)	0.64	(0.61–0.68)
	Hospice	1.05	(0.96–1.14)	0.97	(0.90–1.05)	0.95	(0.88–1.03)	1.03	(0.94–1.12)
	Long-term Care Facilities	1.59	(1.30–1.93)	1.35	(1.30–1.40)	1.13	(1.08–1.18)	1.35	(1.09–1.66)
	Home Care	0.44	(0.13–1.43)	1.36	(1.08–1.70)	0.44	(0.13–1.43)	1.22	(0.96–1.54)
	None of the above	1.20	(1.13–1.27)	1.28	(1.25–1.31)	1.17	(1.10–1.24)	1.23	(1.21–1.26)
	Total	1.07	(1.04–1.11)	1.16	(1.15–1.18)	0.97	(0.94–1.00)	1.02	(1.00–1.03)

## Data Availability

A license for the use of the data has been granted by the Regional Regulation for personal data but restrictions apply; therefore, the data used for the analysis cannot be made publicly available under current rules.

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
