# Peer review of "General and COVID-19-Related Mortality by Pre-Existing Chronic Conditions and Care Setting during 2020 in Emilia-Romagna Region, Italy"

_ijerph, 2021, doi:10.3390/ijerph182413224_

Round 1
Reviewer 1 Report
General and COVID-19-related mortality by pre-existing chronic conditions and care setting during 2020 in Emilia-Romagna region, Italy
Review Report
This paper aims at quantifying excess mortality variability over time and in relation to demographic and pre-existent chronic conditions and care setting in Emilia-Romagna region (Northern Italy). This is a registry-based study comparing the 2020 observed mortality with figures of the previous 5 years by age, sex, month, place of death, and pre-existing chronic conditions and including 300,094 deaths referred to 18+ aged people resident in Emilia-Romagna region. In summary, this is an interesting topic. However, this version needs a minor revision to improve the overall standard of the article.
- The authors consider 5 main factors related to mortality: age, sex, month, place of death and pre-existing chronic conditions. Why choose these five factors? Are there interactions among these explanatory variables?
- Introduce the Mortality Rate Ratios and Prevalence Ratios. How to estimate these two criteria?
- In subsection 2.3.3, the authors present statistical analysis methods used in this paper. The presentation is simple. Please provide the detailed model. For example, the formulae for Poisson models and log-binomial models, respectively for estimate of MRR and PR.
- Figure 1 shows number of daily deaths by sex and age classes, and there are many important findings. Please discuss the hidden and deep-seated reasons in these findings.
Reviewer 2 Report
The authors have presented the general and COVID-19-related mortality by pre-existing chronic conditions and care setting during 2020 in the Emilia-Romagna region, Italy. The study brings a significant contribution to the field. The quality of the presentation is high, with a logical flow and scientific soundness.
Minor points:
- The tables and figures need improvements by adding the legend (abbreviation) and statistical tests below each of them.
- Please consider adding specific details and underlying the importance of the results in practice in the conclusion.
Reviewer 3 Report
This is a very interesting article and well written.
I have however several points to be considered before the article is passed to be published:
- Line 18, please delete the blank line,
- Please either provide a reference or explain why: "...thanks to a higher diagnostic capacity..." - - I am not convinced this is the right explanation (see lines 42 and 43),
- Line 55, please correct the reference,
- To objective of the article provided in lines 13-15 is not consistent with the objective provided in lines 57-59,
- "study design" who the authors describe over there is not a study design - please correct
- Are the RHICS and the regional COVID-19 notifications system available online? Can you provide the reference or better describe these databases?
- Line 119 please explain how was the trend estimated, which statistical methods have been applied?
- Section 2.2.3 - please improve the formatting
- Please provide the limitations of the study in the end of discussion section
- Conclusions provided in the abstract (see lines 19-25) are inconsistent with the conclusions provided in lines 284-290
- The conclusion section requires considerable improvements. Please think which of obtained results are the most important to readers and provided them. For this moment this section is to week. If I had to point out the weakest part of the article it would be conclusions section. The authors have done a great job but do not present it in the most important part of the article, i.e. conclusion section, which should directly expand the current knowledge.
Besides of aforementioned points to be improved, which should not take significant time to be done, I consider the article should be published in an impact factor journal, like IJERPH.
